# Matrine Ameliorates DSS-Induced Colitis by Suppressing Inflammation, Modulating Oxidative Stress and Remodeling the Gut Microbiota

**DOI:** 10.3390/ijms25126613

**Published:** 2024-06-16

**Authors:** Ningning Mao, Yaming Yu, Jin He, Yang Yang, Zhenguang Liu, Yu Lu, Deyun Wang

**Affiliations:** 1Institute of Traditional Chinese Veterinary Medicine, College of Veterinary Medicine, Nanjing Agricultural University, Nanjing 210095, China; 2021207065@stu.njau.edu.cn (N.M.); hnyuym@stu.njau.edu.cn (Y.Y.); 2021207042@stu.njau.edu.cn (J.H.);; 2MOE Joint International Research Laboratory of Animal Health and Food Safety, College of Veterinary Medicine, Nanjing Agricultural University, Nanjing 210095, China; 3Institute of Veterinary Immunology & Engineering, Jiangsu Academy of Agricultural Sciences, Nanjing 210014, China

**Keywords:** DSS, matrine, ulcerative colitis, Treg/Th17, gut microbiota

## Abstract

Matrine (MT) possesses anti-inflammatory, anti-allergic and antioxidative properties. However, the impact and underlying mechanisms of matrine on colitis are unclear. The purpose of this research was to examine the protective impact and regulatory mechanism of matrine on dextran sulfate sodium (DSS)-induced ulcerative colitis (UC) in mice. MT alleviated DSS-induced UC by inhibiting weight loss, relieving colon shortening and reducing the disease activity index (DAI). Moreover, DSS-induced intestinal injury and the number of goblet cells were reversed by MT, as were alterations in the expression of zonula occludens-1 (ZO-1) and occludin in colon. Simultaneously, matrine not only effectively restored DSS-induced oxidative stress in colonic tissues but also reduced the production of inflammatory cytokines. Furthermore, MT could treat colitis mice by regulating the regulatory T cell (Treg)/T helper 17 (Th17) cell imbalance. We observed further evidence that MT alleviated the decrease in intestinal flora diversity, reduced the proportion of *Firmicutes* and *Bacteroidetes*, decreased the proportion of *Proteobacteria* and increased the relative abundance of *Lactobacillus* and *Akkermansia* in colitis mice. In conclusion, these results suggest that MT may mitigate DSS-induced colitis by enhancing the colon barrier integrity, reducing the Treg/Th17 cell imbalance, inhibiting intestinal inflammation, modulating oxidative stress and regulating the gut microbiota. These findings provide strong evidence for the development and application of MT as a dietary treatment for UC.

## 1. Introduction

Ulcerative colitis (UC), which spreads from the rectum through to the colon, is a recurrent inflammatory bowel disease (IBD) caused by intestinal dysfunction [1]. The main symptoms of UC include diarrhea, constipation and bowel discomfort, which seriously reduce the quality of life [2,3]. Studies have shown that the decreased expression of tight-junction proteins in ulcerative colitis may further lead to the development of colorectal cancer [4]. Over the last few decades, the incidence of UC has increased globally, especially in newly industrialized countries [5]. Environmental factors, dysfunction of the immune response, a damaged intestinal epithelial barrier, genetic factors and intestinal microbial disorders could be involved in the occurrence and development of UC [6]. Clinically, immunosuppressants, anti-inflammatory drugs, glucocorticoids and amino salicylates are mainly used to ameliorate UC [7]. Nevertheless, these pharmaceutical interventions have exhibited numerous limitations and side effects, and their efficacy may not extend to all patient populations [8]. For these reasons, it is urgent to explore novel therapeutic strategies and to develop cost-effective formulations for UC patients [9]. Recently, drug development strategies based on natural products from medicinal plants have been considered a promising approach for UC prevention due to their widespread biological activities and good biosafety [10].

Dysfunction of the intestinal barrier is related to the occurrence and progression of UC [11]. An integral gut barrier could maintain intestinal homeostasis by blocking the entry of toxins and pathogens [12]. In addition, there is increasing evidence that the gut microbiota is involved in the pathogenesis of UC [13]. When the balance of the gut microbiota is disturbed, harmful bacteria disrupt the intestinal barrier, leading to the invasion of microbes and toxins into the gut, further exacerbating intestinal inflammation [14]. Therefore, maintaining the integrity of the intestinal barrier and remodeling the structure of the intestinal flora are crucial to the treatment of UC.

Matrine (whose structure is shown in Figure 1), a tetracyclo-quinolizidine alkaloid, is isolated from *Sophora flavescens* by ethanol and other organic solvents [15]. Due to its wide range of biological activities and agricultural properties, research on the phytochemistry, pharmacology and mechanisms of action of matrine has received much attention [16]. Matrine has anti-inflammatory, antioxidative and neuroprotective pharmacological activities, and it can easily penetrate the biofilm barrier to exert pharmacological effects [17]. Research has demonstrated that matrine can reduce apoptosis and protect the integrity of mucosal epithelial cells [18]. Another study suggested that matrine could palliate the symptoms of cecal ligation and puncture (CLP)-induced sepsis in mice and restrain NLRP3 inflammasome activation by regulating the PTPN2/JNK/SREBP2 signaling pathway [19]. Additionally, matrine can ameliorate colitis by regulating the intestinal flora [20]. UC mice induced by DSS are a typical model used to simulate human UC and have been widely used in intestinal health studies [21]. However, the effects and mechanism of matrine on DSS-induced UC remain unexplored.

Therefore, the present study established a DSS-induced UC model and used it to demonstrate the ameliorative effect of matrine on DSS-induced colitis in mice and to examine its regulatory mechanism. This mainly highlighted the modulatory effect of matrine on the inflammatory response, intestinal barrier and intestinal flora, and elucidated the potential mechanisms of its various modulatory effects, providing a detailed theoretical basis for the alleviation of colitis by matrine.

## 2. Results

### 2.1. Matrine Alleviated DSS-Induced Colitic Symptoms

To determine the effect of matrine on colitis, mice were fed 3% DSS to construct a UC model, and then the effect of matrine on colitis symptoms was evaluated (Figure 2A). As shown in Figure 2, weight loss and bloody stools were observed after oral administration of DSS as compared with the CON group. However, weight loss was reversed after treatment with matrine (Figure 2B), while the disease activity index (DAI) score was significantly reduced (*p* < 0.05, Figure 2C).

It is commonly believed that colonic contraction is a major symptom of UC, and recurring instances of inflammation could result in colonic edema [23]. The results showed that mice in the DSS group had shortened colons, developed ulcerated edema, and had significantly increased colon weight/colon length ratios compared with the CON group (*p* < 0.05, Figure 2D–F). However, matrine inhibited the colonic shortening and damage in colitis mice (*p* < 0.05, Figure 2D–F). In conclusion, these results showed that matrine could induce remission of the DSS-induced experimental colitis in mice.

### 2.2. Histopathologic Analysis of the Colon

Next, we evaluated the effects of matrine administration on DSS-induced colonic tissue injury in mice. Hematoxylin–eosin (H&E) staining showed that DSS caused severe damage to the colonic tissues of mice, with the swelling and destruction of crypts and infiltration of inflammatory cells. However, oral administration of matrine significantly reduced colon damage; this included having an improved histological structure, a more complete colitis mucosal epithelium, and reduced inflammatory cell infiltration, and most goblet cells maintained a normal physiological morphology (Figure 3A,B). Furthermore, a significant reduction in goblet cells was observed in the DSS group. In contrast, matrine mitigated damage to the crypts and epithelial cells and resulted in an abundance of goblet cells (*p* < 0.05, Figure 3C,D). In conclusion, matrine treatment significantly improved colonic injury.

### 2.3. Matrine Alleviated DSS-Induced Colonic Inflammation

Cytokines play a key role in regulating inflammation, which is an essential effector in the progression of colonic inflammation and can be used to show the extent of colonic damage [24]. To further assess the impacts of matrine against colonic inflammation, the enzyme-linked immunosorbent assay (ELISA) and reverse transcription–polymerase chain reaction (RT-qPCR) tests were performed, respectively. In this study, an inflammatory response appeared in the DSS group, as evidenced by the significant increase in tumor necrosis factor-α (TNF-α), interleukin-6 (IL-6), IL-17A, IL-1β and nitric oxide (NO) and the remarkable decrease in IL-10 (*p* < 0.05, Figure 4A–F). However, administering matrine to colitis mice effectively suppressed the intestinal inflammatory response (*p* < 0.05). The results of the RT-qPCR showed that the expression of cytokine genes was consistent with the results from the ELISA (Figure 4G–L).

### 2.4. Matrine Alleviated DSS-Induced Colonic Oxidative Stress

During the UC process, oxidative stress leads to changes in the activity of oxidative-stress-related enzymes [25]. In order to investigate the effects of matrine on oxidative stress, the level of malondialdehyde (MDA), superoxide dismutase (SOD) and catalase (CAT) and the total antioxidant capacity (T-AOC) were measured. As depicted in Figure 5, oxidative stress was exacerbated in the DSS group, leading to tissue damage (*p* < 0.05). However, the T-AOC and SOD and CAT activities of colonic tissues increased significantly after oral administration of matrine (Figure 5B–D), while MDA levels decreased significantly (*p* < 0.05, Figure 5A).

### 2.5. Matrine Protected against DSS-Induced Intestinal Barrier Damage

Tight junctions (TJs) are key proteins that regulate intestinal permeability and maintain gut barrier function [26]. TJs are composed of a family of transmembrane proteins (claudin, occludin) and scaffolding proteins (ZO-1) [27]. When the intestinal barrier is impaired, expression of TJs is reduced in intestinal epithelial cells [28]. To assess the protective effect of matrine on the intestinal barrier, the expression of the ZO-1 and occludin proteins was assayed. As demonstrated in Figure 6, DSS intervention disrupted the epithelial barrier and reduced the distribution of ZO-1 and occludin. Conversely, matrine increased the expression of these proteins in colonic tissues and decreased the permeability of the intestinal mucosa (Figure 6B–C). The results showed that matrine protected against DSS-induced colitis by upregulating the expression of TJ proteins in colonic tissues.

### 2.6. Matrine Regulated the Balance of Treg/Th17 Cells in Colitis Mice

There is a dynamic balance between Treg and Th17 cells, and an imbalance between Treg and Th17 cells may affect UC progression [29]. The results showed that a typical Treg/Th17 cell imbalance was presented in the DSS group (*p* < 0.05, Figure 7). The proportion of CD4+ IL-17A+ cells in mesenteric lymph nodes (MLNs) and Peyer’s patches (PPs) of the DSS group was significantly higher than that in the CON group, showing an exacerbated inflammatory response, while this was downregulated following the administration of matrine (*p* < 0.05). In addition, matrine treatment increased the ratio of CD4+ Foxp3+/CD4+ IL-17A+ cells (*p* < 0.05). In conclusion, matrine inhibited the inflammatory response by modulating the ratio of Th17 and Treg cells in colitis mice (*p* < 0.05).

### 2.7. Matrine Modulated the Intestinal Flora in Colitis Mice

Based on the above findings, we focused on the mid-dose group to determine the effects of matrine on the gut microbiota by 16S ribosomal RNA (16S rRNA) gene sequencing. The alpha diversity showed no significant differences in the Chao1 and eabundance-based coverage estimator (ACE) indexes among all the groups, but the Shannon index was significantly decreased in the DSS group (*p* < 0.05, Figure 8A–F). Nevertheless, the treatment group increased the diversity of the gut microbiota. The β-diversity of gut microbiota composition between the different groups was analyzed by principal coordinate analysis (PCoA). The structure of the gut microbiota community was significantly different between the DSS and CON groups (*p* = 0.0001), whereas the MT-M group was more similar to the CON group (*p* = 0.0079, Figure 8G). The taxonomic cladogram (Figure 8H) and the linear discriminant analysis (LDA) score (Figure 7I) showed that DSS treatment upregulated the abundance of *Turicibacter*, *Romboutsia*, *Peptostreptococcaceae*, *Mucispirillum*, *Deferribacterales*, *Deferribacteraceae* and *Deferribacteres*.

At the phylum level, the predominant phyla were *Firmicutes* and *Bacteroidetes*. The relative abundance of *Bacteroidetes* decreased and the ratio of *Firmicutes* to *Bacteroidetes* increased after DSS stimulation (*p* < 0.05, Figure 9C,D). However, the opposite trend appeared in the MT-M group, where matrine treatment increased the abundance of *Bacteroidetes* (*p* < 0.05). In addition, matrine reduced the relative abundance of *Proteobacteria* compared with the DSS group (*p* < 0.05, Figure 9E), suggesting that matrine had a regulatory effect on specific microbial communities. The abundance of intestinal flora by genus was further analyzed. As shown in Figure 9F, *Lactobacillus*, *Akkermansia*, *Turicibacter*, *Saccharibacteria_genera_incertae_sedis*, *Anaerobacterium*, *Eisenbergiella*, *Helicobacter*, *Bacteroides*, *Clostridium_XlVa* and *Enterorhabdus* are the most abundant bacteria groups. The intestinal *Lactobacillus* and *Akkermansia* abundance in the DSS group was significantly lower, whereas matrine increased the abundance of *Lactobacillus* and *Akkermansia* (*p* < 0.05, Figure 9G–H). Overall, these data suggest that matrine attenuates DSS-induced colitis by remodeling the gut microbiota.

## 3. Discussion

UC is one of the major forms of inflammatory bowel disease (IBD), and the clinical diagnosis of UC has increased with the improvement of modern medical diagnostic and treatment techniques [30]. UC has become a huge burden on human life due to its recurring course, difficulty in curing and tendency to deteriorate into cancer. Thus, there is an imminent need to develop affordable, effective and sustainable treatments [31]. Matrine, an active ingredient extracted from the natural product *Sophora flavescens*, has been found have the potential to improve intestinal health and to interfere with the development of colitis [20]. However, the mechanisms of matrine in regulating the intestinal flora and apoptosis signaling pathways need to be further explored.

Mice taking DSS orally have been reported to exhibit severe colitis features similar to those in human UC patients [32]. In the present study, mice in the model group showed symptoms such as a shortened colon, weight loss and thinning stool, indicating that the colitis model was successfully established. However, oral matrine significantly improved these indicators. HE and AB-PAS staining showed that the typical images observed in the DSS group were a disturbed colonic structure, marked inflammatory cell infiltration, a reduction in goblet cells and mucosal ulceration. However, matrine restored colonic mucosal damage and prevented goblet cell loss, thereby improving UC symptoms.

UC is accompanied by an abnormal immune response, and cytokines play a key role in its disease management [33,34]. Inappropriate elevation of inflammatory cytokines is an indicator of the severity of colitis and could be used to show the extent of colonic damage [35]. The important role of cytokines in immunomodulation and inflammation has been demonstrated, and drugs targeting cytokines are considered promising candidates for the treatment of UC [36]. Matrine treatment reduced the secretion of TNF-α, IL-6, IL-1β, IL-17A and NO in colitis mice, inhibiting the colon inflammatory response. These cytokines interact to disrupt the epithelial barrier and mediate an inflammatory response [37]. These data suggest that the anti-inflammatory effect of matrine is due to its ability to reduce the overexpression of inflammatory cytokines in DSS-induced mice.

During UC, patients exhibit lower antioxidative capacity, and they show higher levels of oxidative DNA damage than healthy individuals [38]. Administration of DSS caused oxidative damage to the gut, as evidenced by reduced levels of the antioxidative enzymes. After oral administration of matrine, the T-AOC and the content of SOD, CAT and MDA in the colon was restored to normal. These results suggest that matrine may relieve colitis by regulating the intestinal oxidative stress levels.

TJPs are crucial components for maintaining the integrity and normal function of the gut barrier [39]. The disruption of TJPs would increase the intestinal permeability and the inflammatory response [40]. Studies have shown that DSS treatment enhances the intestinal permeability and downregulates TJ protein expression [41]. However, matrine treatment reduced the loss of occludin and ZO-1 proteins, indicating that matrine contributed to the restoration of barrier integrity disrupted by DSS. In addition, it has been reported that natural extracts could prevent colitis by increasing the expression of TJPs [42], which is consistent with our result.

Th17/Treg imbalance causes hyperactivation of the immune system, which, in turn, leads to an inflammatory response [43,44]. In experiments, the number of Th17 cells was increased in colitis mice, whereas matrine reversed the Treg/Th17 imbalance in MLNs and PPs of colitis mice. It is reported that berberine treatment could alleviate inflammatory bowel disease by improving the Treg/Th17 balance [45], which is consistent with our findings. Together, these results provide evidence that matrine prevents UC by regulating the Treg/Th17 imbalance.

More and more evidence shows that the gut microbiota is closely related to the pathogenesis of UC, which can resist pathogens and regulate the immune response [46]. Adjusting the structure of the intestinal flora may be a treatment method for UC. In this study, matrine treatment could increase the diversity of the microbial community. The elevated Shannon index in the treatment group suggests an increase in the abundance of gut microbiota. The β-diversity analysis showed that matrine could alter the overall microbial community composition in UC mice. These results show that matrine can alleviate the DSS-induced gut microbiota disorder and decrease in diversity in mice.

To further investigate the changes in gut microbiota composition after the administration of matrine, we analyzed the bacterial community composition by phylum and genus. At the phylum level, oral matrine significantly reduced the proportion of *Firmicutes* and *Bacteroidetes* in colitis mice. It has been reported that *Firmicutes* mainly removes complex carbohydrates, while *Bacteroidetes* tend to remove oligosaccharides [47]. *Proteobacteria* include many pathogenic bacteria that are responsible for the overproduction of pro-inflammatory cytokines. Thus, increased amounts of *Proteobacteria* may induce a variety of intestinal disorders. Reduction of *Proteobacteria* in the treatment group indicated that matrine could reduce the levels of harmful intestinal bacteria. At the genus level, matrine increased the relative abundance of *Lactobacillus* and *Akkermansia* in the colitis mice. As the first generation of probiotics, *Lactobacillus* could enhance the intestinal barrier function and provide resistance to pathogens [48]. *Akkermansia* is associated with intestinal health and plays an important role in the mucosal surface of the host cavity. It has been reported that *Akkermansia* maintains intestinal homeostasis by binding to Toll receptors in colon epithelial cells, thereby improving obesity and IBD [49]. These results suggest that matrine could regulate the multifariousness and composition of intestinal flora, bring more benefits to the host.

## 4. Materials and Methods

### 4.1. Reagents

Matrine was purchased from Shanghai Maclin Biochemical Technology Co., Ltd., Shanghai, China (Lot No. M813524). DSS (36,000–50,000 MW) was purchased from Dalian Meilun Biotechnology Co., Ltd., Dalian, China (Lot No. J0712B). TNF-α (EK282P/4), IL-1β (EK201BP), IL-10 (EK210P), IL-17A (EK217/2) and IL-6 (EK206P) cytokine ELISA kits were purchased from MultiSciences Biotech Co., Ltd. (Hangzhou, China). The NO (S0021S) kit was purchased from Beyotime Biotech Co., Ltd. (Shanghai, China). Primers and TRIzol reagents used in this study were purchased from Nanjing Qingke Biotechnology Co., Ltd. (Nanjing, China). The HiScript^®^ II RT SuperMix and ChanmQTM SYBR^®^ qPCR Master Mix were bought from Vazyme Biotech Co., Ltd. (Nanjing, China). Anti-CD4 (E-AB-F1097UC), anti-IL-17A (E-AB-F1272UD), and anti-Foxp3 (E-AB-F1238E) antibodies were purchased from Elabscience Biotechnology Co., Ltd., (Wuhan, China). All other reagents were analytical grade.

### 4.2. Mice

C57BL/6J mice (6 weeks old) were bought from the Animal Research Center of Yangzhou University (Yangzhou, Jiangsu, China) (Permission No. SYXK (Jiangsu) 2022–0009). Animal experiments were conducted in accordance with the Nanjing Agricultural University Laboratory Animal Center Animal Ethics Committee approval number: NJAUPZ2022-0611.

### 4.3. Animal Experiment Design

C57BL/6J mice were fed for one week before being split into six groups: the control group (CON), model group (DSS), positive control group (5-ASA) and the 10 mg/kg matrine (MT-L), 20 mg/kg matrine (MT-M) and 30 mg/kg matrine (MT-H) treatment groups (6 mice per group). Except for the control group, which was given pure water during the experiment, the other five groups were given 3% DSS solution orally for seven days. Then, the experimental groups were each given different doses of matrine once a day. The positive group (5-ASA group) was administered 5-aminosalicylic acid (200 mg/kg) (Shanghai Maclin Biochemical Co., Ltd., Shanghai, China) intragastrically. Other groups were treated with 0.9% NaCl solution by oral gavage for 7 days. The DAI score was assessed by body mass index, stool shape and bleeding. After the experiment, the colon tissues were collected, photographed and measured. The cecum contents were collected and stored at −80 °C.

### 4.4. Colon Histology

The collected colon tissues were fixed, embedded in paraffin and stained with hematoxylin and eosin after dewaxing. Histological analysis was evaluated in a blinded manner [50]. The microscope (Leica, Tokyo, Japan) was used to capture the images. The slices were scored according to the degree of inflammatory infiltration (0–5), the degree of indentation damage (0–4), the degree of ulceration (0–3), and the presence of edema (0 or 1).

### 4.5. Alcian Blue–Periodic Acid Schiff (AB-PAS) Staining

For further histological evaluation, AB-PAS staining was performed [51]. Briefly, the deparaffinized sections were immersed in Alcian Blue dye for 10 min, oxidized with 0.5% periodate aqueous solution after washing, then dyed with Schiff’s reagent in a dark place for 30 min. A fluorescent microscope (Leica, Tokyo, Japan) was used to capture the images.

### 4.6. Cytokine Analysis of Colonic Tissues

The colon samples were ground with PBS (contain PMSF) and centrifuged to obtain colonic tissue homogenates. Levels of the cytokines IL-6, IL-1β, IL-10, IL-17A, NO and TNF-α in the mice colon were determined by ELISA kits (MultiSciences, China).

### 4.7. Quantitative Real-Time PCR Analysis

Total RNA was extracted from colon tissues using the TRIzol reagent and reversed transcribed into cDNA [52]. Expression levels were quantified by qRT-PCR analysis with the BIO-RAD CFX Connect Real-Time system. The primers used in the experiment are listed in Table 1. The relative amount of each gene was calculated by the 2^−∆∆Ct^ method, and β-actin was used as the internal reference gene.

### 4.8. Evaluation of Oxidative Stress

Colon tissues were homogenized in the manufacturer’s buffer and the T-AOC and levels of CAT, SOD and MDA were detected using assay kits according to the manufacturer’s instructions.

### 4.9. Immunofluorescence Staining

The expression of tight-junction proteins (TJPs) in the colon were measured by immunofluorescence staining [53]. Briefly, the tissue sections were first dewaxed and incubated for 1 h with a buffer composed of 10% goat serum, 1% bovine serum albumin (BSA) and 0.5%Triton X-100. After that, ZO-1 and occludin antibodies were incubated with the colon sections, and the corresponding fluorescent secondary antibodies (ZO-1-cy3 and Occludin-alexa488) were added and the signal allowed to develop for 2 h. A fluorescent microscope (Leica Microsystems, Wetzlar,, Germany) was used to capture the images.

### 4.10. Flow Cytometry

Mesenteric lymph nodes (MLNs) and Peyer’s patches (PPs) were collected, and then the cell suspensions were collected by mechanical grinding and filtration (70 μm sieve). After that, cells were collected through centrifugation and then re-suspended in PBS (1 × 10^6^ cells/mL). Cells were incubated with anti-CD4 antibody for 30 min in the dark, followed by fixation and permeabilization for 40 min. Next, intracellular staining was performed with the anti-IL-17A antibody and anti-Foxp3 antibody. Finally, samples were analyzed by FACS Canto II flow cytometry. Data were analyzed using Flow Jo software V10.

### 4.11. 16S rRNA Sequencing

Total genomic DNA of gut microbiota was extracted using the fecal Fast DNA SPIN Kit (MP Biomedicals, Santa Ana, CA, USA). The V3–V4 regions of the bacterial 16S DNA gene were amplified and sequenced using the Illumina Novaseq 6000 sequencer. Subsequently, a sequencing library was constructed using the MiSeq Reagent Kit v3 (Illumina, CA, USA) and then sequenced using an Illumina MiSeq Benchtop Sequencer (Illumina). Finally, the previous bioinformatics analysis method was used for sequence analysis [54].

### 4.12. Statistical Analysis

Statistical calculations were analyzed by GraphPad Prism version 8.0 software, and statistical significance was analyzed by Duncan’s multiple-range tests. The results are presented as the mean ± SEM, and * *p* < 0.05, ** *p* < 0.01 and *** *p* < 0.001 were considered statistically significant.

## 5. Conclusions

In summary, the present study demonstrated the palliative effect of matrine on colitis. Although its specific mechanism of action still needs to be further explored, it is worth noting that matrine could inhibit inflammatory responses, regulate intestinal oxidative stress levels, improve the intestinal barrier integrity, upregulate the Treg/Th17 ratio and play a powerful therapeutic role in UC. Finally, matrine could regulate key components of the gut microbiota. Overall, matrine may be a promising treatment for UC.

## Figures and Tables

**Figure 1 ijms-25-06613-f001:**
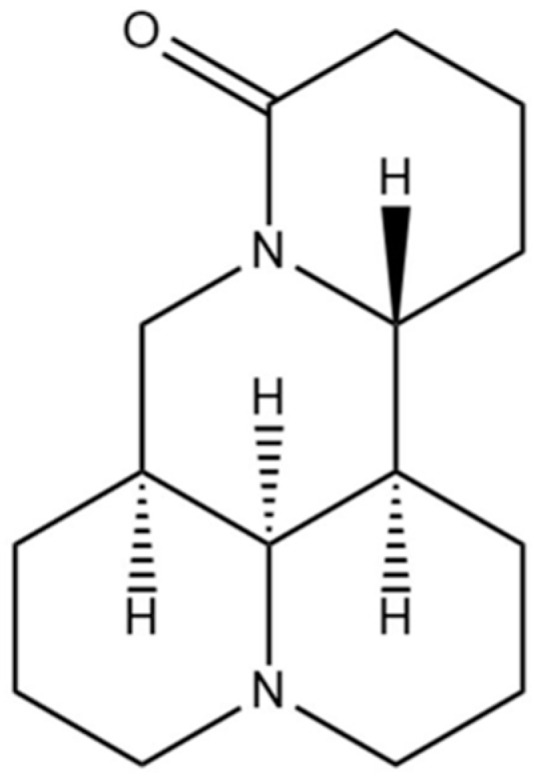
The chemical structure of matrine [22].

**Figure 2 ijms-25-06613-f002:**
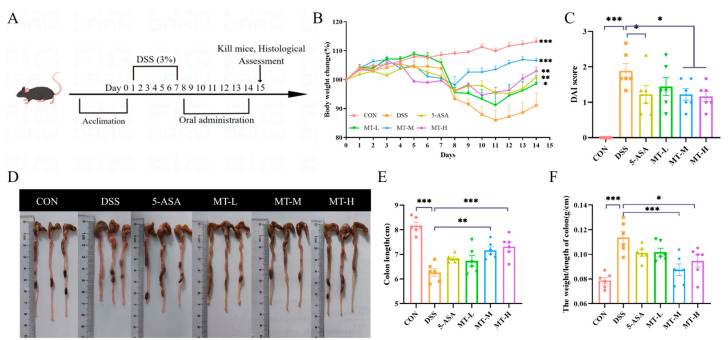
Matrine alleviated DSS-induced colitis in mice. (**A**) Schematic of the experimental design. (**B**) Daily body weight changes during the experiment. (**C**) Disease activity index (DAI). (**D**) Representative photo of the colon in each group and (**E**) quantitative analysis of the colon length in each group. (**F**) Colonic weight/colonic length. Data are expressed as means ± SEM; *n* = 6 for each group. * *p* < 0.05, ** *p* < 0.01 and *** *p* < 0.001 vs. the DSS group.

**Figure 3 ijms-25-06613-f003:**
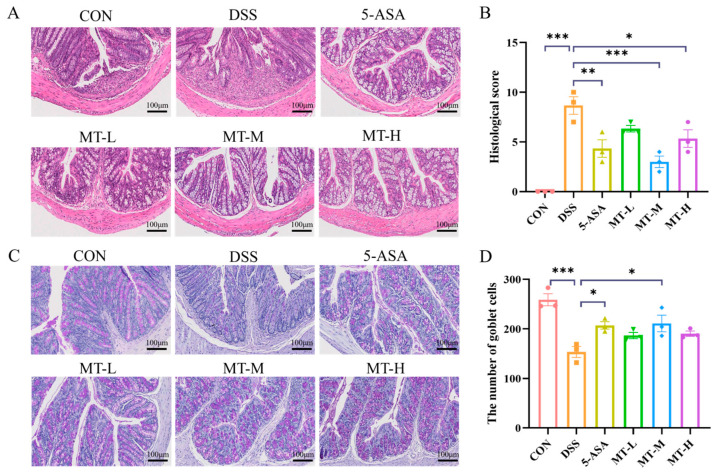
Histopathological analysis of the colon. (**A**) Representative H&E staining of colonic tissues. (**B**) Histopathological scores of colons in each group. (**C**) Alcian Blue–periodic acid Schiff (AB-PAS) stain. (**D**) The number of goblet cells. Scale bar: 100 μm; original magnification: 200×. Data are expressed as means ± SEM; *n* = 3 for each group. * *p* < 0.05, ** *p* < 0.01 and *** *p* < 0.001 vs. the DSS group.

**Figure 4 ijms-25-06613-f004:**
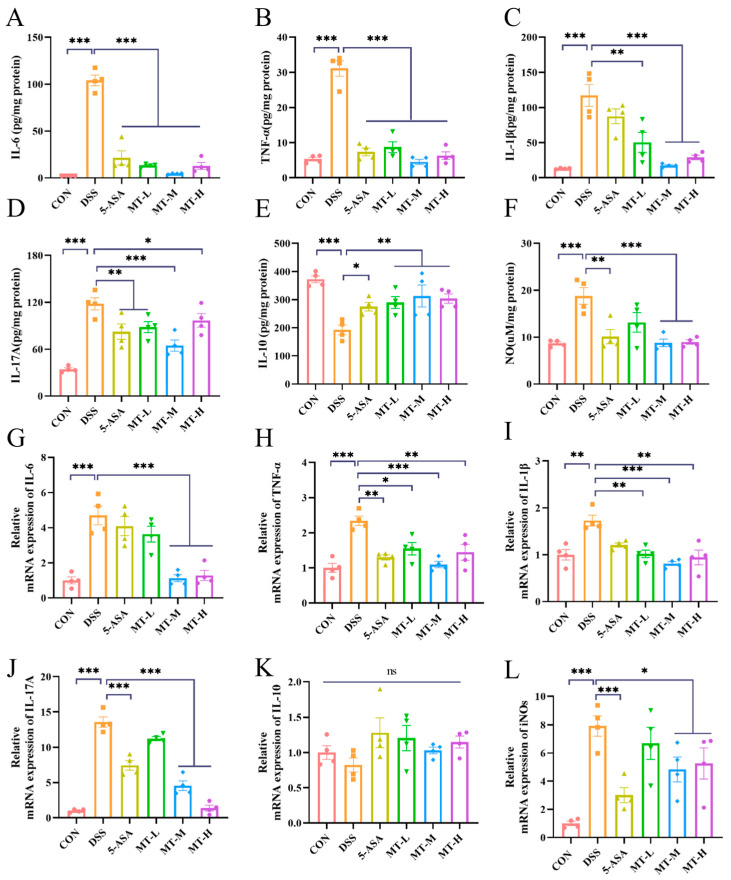
Matrine ameliorated the secretion of cytokines and the expression of related genes in colonic tissues. Colonic tissue inflammatory protein levels: (**A**) IL-6, (**B**) TNF-α, (**C**) IL-1β, (**D**) IL-17A, (**E**) IL-10 and (**F**) NO. mRNA expression levels of (**G**) IL-6, (**H**) TNF-α, (**I**) IL-1β, (**J**) IL-17A, (**K**) IL-10 and (**L**) iNOs in the colon of colitis mice were determined by RT-qPCR. Data are expressed as means ± SEM; *n* = 4 for each group. * *p* < 0.05, ** *p* < 0.01 and *** *p* < 0.001 vs. the DSS group, and ns indicates no significant difference between groups.

**Figure 5 ijms-25-06613-f005:**
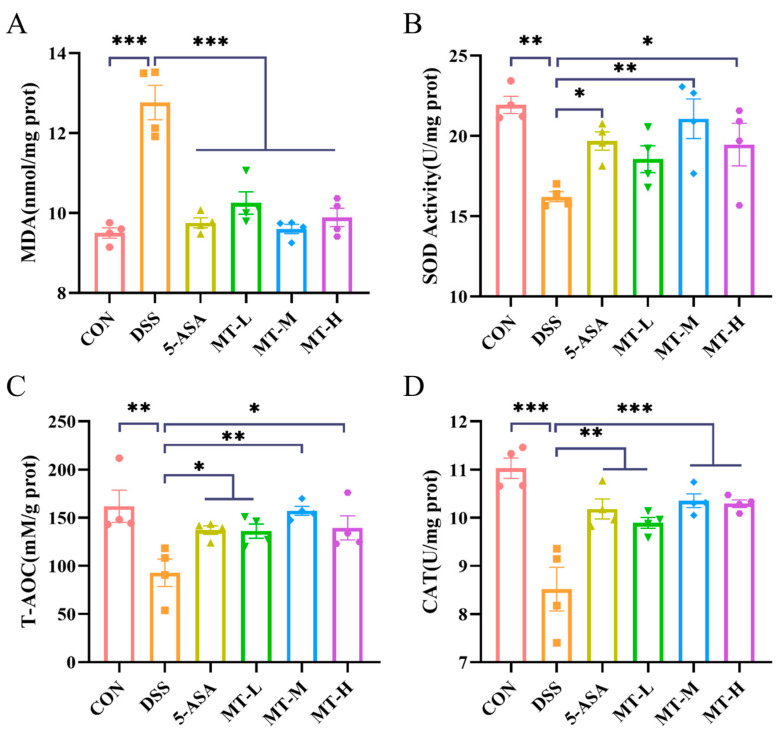
Matrine alleviated oxidative stress in colitis mice. (**A**) MDA, (**B**) SOD, (**C**) T-AOC, (**D**) CAT levels. Data are expressed as means ± SEM; *n* = 4 for each group. * *p* < 0.05, ** *p* < 0.01 and *** *p* < 0.001 vs. the DSS group.

**Figure 6 ijms-25-06613-f006:**
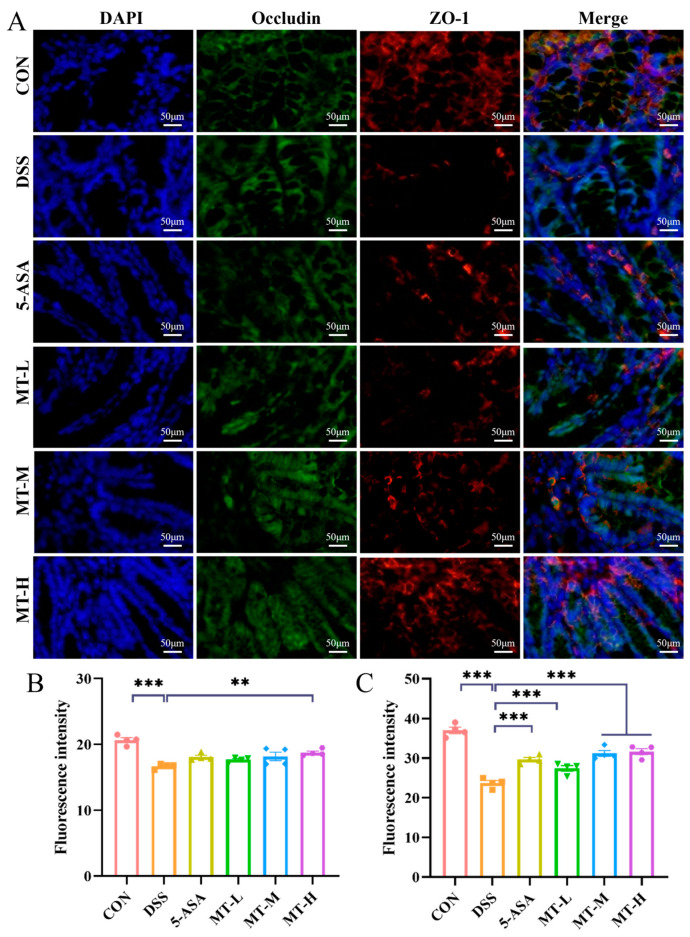
Immunofluorescence analysis of the ZO-1 and occludin proteins in the colon. (**A**) The nucleus, occludin and ZO-1 were stained using DAPI (blue), Alexa488 (green), and CY3 (red), respectively. (**B**) Relative density of occludin-positive regions. (**C**) Relative density of ZO-1-positive regions. Data are expressed as means ± SEM; *n* = 4 for each group. Scale bar: 50 μm; original magnification: 400×. * *p* < 0.05, ** *p* < 0.01 and *** *p* < 0.001 vs. the DSS group.

**Figure 7 ijms-25-06613-f007:**
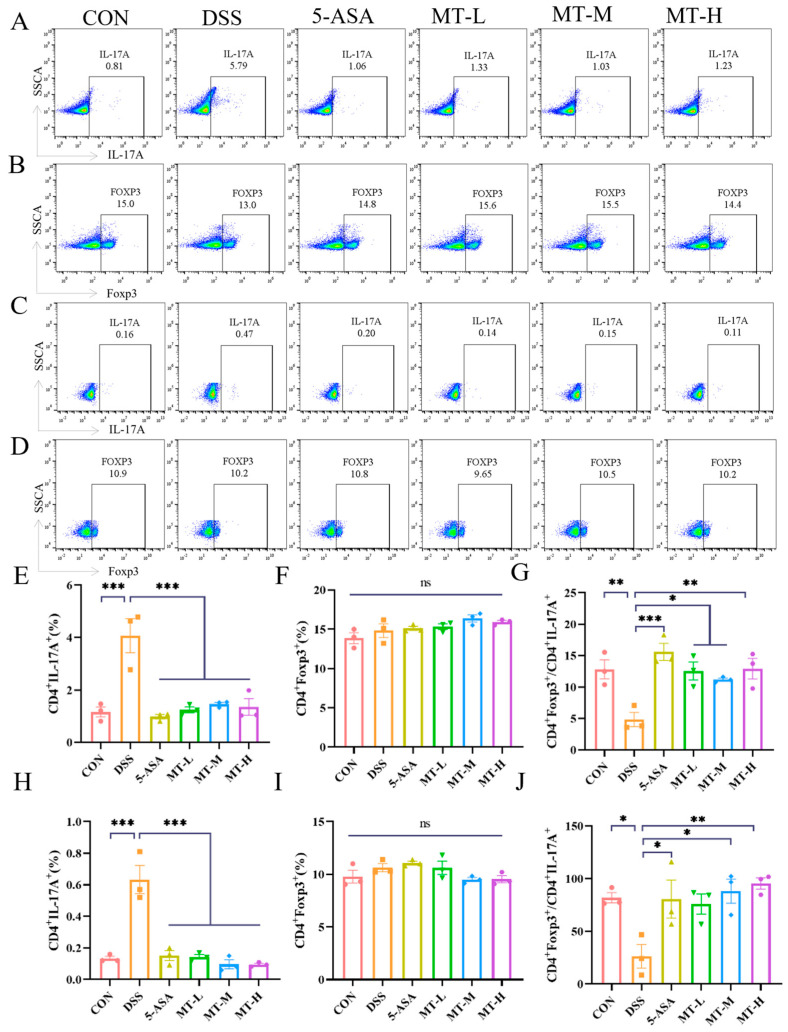
Matrine regulated the balance of Treg/Th17 cells in colitis mice. (**A**) Representative experiment of the ratio of CD4+ IL-17A+ Th17 cells in the MLNs as determined by flow cytometry. (**B**) Representative experiment of the ratio of CD4+ Foxp3+ Treg cells in the MLNs as determined by flow cytometry. (**C**) Representative experiment of the ratio of CD4+ IL-17A+ Th17 cells in PPs as determined by flow cytometry. (**D**) Representative experiment of the ratio of CD4+ Foxp3+ Treg cells in the PPs as determined by flow cytometry. (**E**–**J**) Bar charts showing the expression of IL-17A, Foxp3 and Foxp3/IL-17A. Data are expressed as means ± SEM; *n* = 3 for each group. * *p* < 0.05, ** *p* < 0.01 and *** *p* < 0.001 vs. the DSS group, and ns indicates no significant difference between groups.

**Figure 8 ijms-25-06613-f008:**
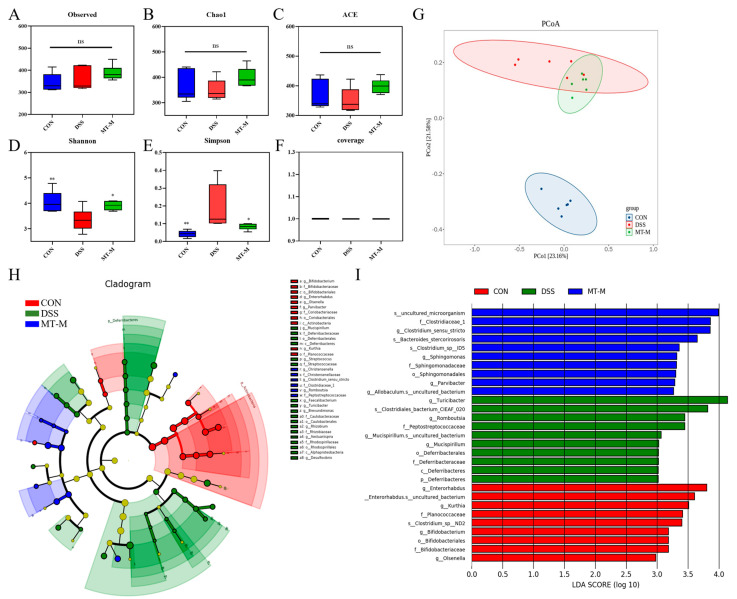
Matrine modulated the composition of gut microbiota in colitis mice. (**A**) Observed species indexes, (**B**) Chao1, (**C**) ACE, (**D**) Shannon, (**E**) Simpson indexes and the coverage (**F**) were used to estimate the alpha diversity among each group to reflect the alpha diversity of the gut microbiota. (**G**) β-diversity analysis of the gut microbiota PCoA analysis based on weighted UniFrac distances. (**H**) LefSe analysis (**I**) LDA analysis. Data are expressed as means ± SEM; *n* = 6 for each group. * *p* < 0.05, ** *p* < 0.01 and *** *p* < 0.001 vs. the DSS group, and ns indicates no significant difference between groups.

**Figure 9 ijms-25-06613-f009:**
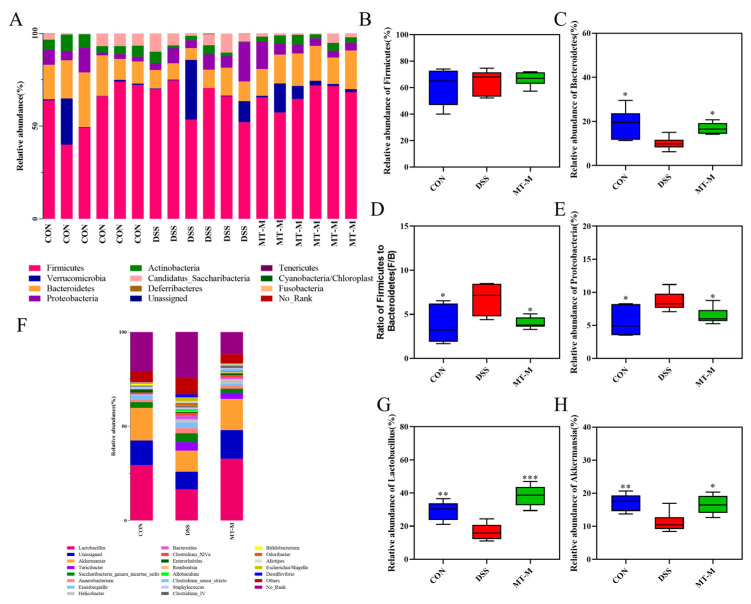
Matrine improved the gut microbiota disorder caused by DSS-induced colitis in mice. (**A**) Bacterial taxonomic profiling at the phylum level. (**B**) Matrine moderated the abundance of *Firmicutes* and (**C**) *Bacteroidetes* and (**D**) the ratio of F/B. (**E**) Matrine decreased the proportion of *Proteobacteria* in the gut microbiota. (**F**) Bacterial taxonomic profiling at the genus level. (**G**) The relative abundance of *Lactobacillus* and *Akkermansia* (**H**). Data are expressed as means ± SEM; *n* = 6 for each group. * *p* < 0.05, ** *p* < 0.01 and *** *p* < 0.001 vs. the DSS group.

**Table 1 ijms-25-06613-t001:** Primer sequences used for the RT-qPCR.

Gene	Forward Primer	Reverse Primer
β-actin	GGGCCGTATTCCCGAGTATC	TTTTGGACTGCGCCTCATCT
IL-1ß	TGCCACCTTTTGACAGTGATG	TTCTTGTGACCCTGAGCGAC
iNOs	CATTCAGATCCCGAAACGCT	TGTAGGACAATCCACAACTCGC
TNF-α	CAGGCGGTGCCTATGTCTC	CGATCACCCCGAAGTTCAGTAG
IL-6	CCAAGAGGTGAGTGCTTCCC	CTGTTGTTCAGACTCTCTCCCT
IL-10	ATGCTGCCTGCTCTTACTGACTG	CCCAAGTAACCCTTAAAGTCCTGC
IL-17A	TTTAACTCCCTTGGCGCAAAA	CTTTCCCTCCGCATTGACAC

## Data Availability

The data presented in this study are available in the article.

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
