# Peer review of "Matrine Ameliorates DSS-Induced Colitis by Suppressing Inflammation, Modulating Oxidative Stress and Remodeling the Gut Microbiota"

_ijms, 2024, doi:10.3390/ijms25126613_

Round 1

Reviewer 1 Report

Comments and Suggestions for Authors

In this study authors examined the protective impact of matrine on dextran sulfate sodium (DSS)-induced UC mice and found that  MT alleviated DSS-induced UC by inhibiting weight loss, relieving colon shortening and reducing disease activity index (DAI). Moreover, DSS-induced intestinal injury and the number of goblet cells were reversed by MT, as were alterations in the expression of ZO-1 and occludin in colon. Matrine also reduced the production of inflammatory cytokines regulating Treg/Th17 cell imbalance.

the manuscript is interesting and generally well written. However, there are several points that must be improved. My commented are listed below. 

Lines 33-40: it deserves to be pointed out that ulcerative colitis is an important risk factor for colorectal cancer occurrence (see PMID: 28586045). This is an important point to add since it can further highlight the interesting results obtained by the authors. In fact, the decreased expression of ZO-1 and occludin found in ulcerative colitis model can favor epitelial-mesenchimal transition, one of the first steps of carcinogenesis. 

Line 56Sophora flavescens must be written in italic

Figure 2A and C: Images are too small to appreciate tissue morphology 

 Figure 5A: Please insert higher magnifications

 Figure 7H and I are unreadable 

Figure 9 should be moved in the introduction where is mentioned for the first time

4.6 Cytokine Analysis of Colonic Tissues: ELISA kits product codes must be added 

4.7 Quantitative Real-Time PCR Analysis: The table of the primers used should be moved here and removed from supplementary material  

Lines 372-374: this is a leftover of the journal template, remove

Supplementary Materials: There is no Figure S1 and Video S1 in the supplementary matherial. I think that this is a leftover of the journal template, remove

Abbreviations must be written in full length when mentioned for the first time

Reviewer 2 Report

Comments and Suggestions for Authors

Dear authors,

Thank you for the opportunity to review your manuscript. It was an interesting read, and I believe only some minor edits are necessary before publication can be considered:

1. Please cite your figures in the text. It is unclear where do they belong. Each figure should appear in text after its first mention.

2. Include in the discussion section a short paragraph discussing the potential clinical applications or medical use of your findings. It should be clarified what are the potential benefits.

3. Study limitations should be mention in the discussion section as well. Please do so.

4. If the chemical structure of matrine is not your original creation, then you must cite the source.

5. It seems that you omitted to write anything in the conclusions section, as it was left blank. Please add your conclusions.

6. Also fill the authors contributions.

7. Minor English edits are required.

Regarding the content and presentation of findings, everything seems well written and presented.

Best regards,

The authors

Comments on the Quality of English Language

Minor edits are required. 

Reviewer 3 Report

Comments and Suggestions for Authors

In this study, Mao et al. investigated the effects of matrine on DSS-induced colitis in mice, highlighting its ability to alleviate symptoms by enhancing colon barrier integrity, reducing Treg/Th17 cell imbalance, and modulating gut microbiota. This study suggested Matrine’s potential as a dietary treatment for ulcerative colitis, providing a basis for its development and application. This study presents a meaningful topic and does an extensive experimentation and analysis, but there are some issues with it. Here are some comments on this study:

1.        It is recommended that the authors consistently use “intestinal flora”, “gut microbiome”, and “gut microbiota” in the manuscript.

2.        There were 6 mice in each group, but the number of samples used for histopathologic analysis and oxidative stress analysis was 3 or 4. Were the samples randomly selected?

3.        Line 184 “The structure of the gut microbial community was significantly different”, it would be useful to use the PERMANOVA analysis the differences.

4.        Section 4.3, how were the low, medium, and high concentrations of matrine determined, any reference to other studies?

5.        Section 4.11, it is recommended that the detailed 16S rRNA analysis information be provided.

6.        Section 5 conclusions, if there are no conclusions, please delete it.

7.        Line 395 “ffnancial”, please revise it.

Round 2

Reviewer 1 Report

Comments and Suggestions for Authors

the manuscript has been significantly improved and can be accepted in the present form